# Comparative Clinicopathological Analysis of Oral Focal Mucinosis and Solitary Cutaneous Focal Mucinosis: A Case Series and Literature-Based Analysis

**DOI:** 10.3390/dermatopathology12040038

**Published:** 2025-10-27

**Authors:** Wickramasinghe Mudiyanselage Sithma Nilochana Wickramasinghe, Primali Rukmal Jayasooriya, Balapuwaduge Ranjit Rigobert Nihal Mendis, Tommaso Lombardi

**Affiliations:** 1Department of Oral Pathology, Faculty of Dental Sciences, University of Peradeniya, Peradeniya 20400, Sri Lanka; simywicks@gmail.com (W.M.S.N.W.); primalijaya@dental.pdn.ac.lk (P.R.J.); ranjtrigobert@gmail.com (B.R.R.N.M.); 2Unit of Oral Medicine and Maxillo-Facial Pathology, Department of Surgery, Division of Maxillofacial and Oral Surgery, University Hospital and Faculty of Medicine, 1205 Geneva, Switzerland

**Keywords:** oral focal mucinosis, myxoid stroma, solitary cutaneous focal mucinosis, clinical diagnosis, recurrence

## Abstract

**Background/Objectives:** Oral focal mucinosis (OFM) and solitary cutaneous focal mucinosis (SCFM) are rare, benign lesions characterized by localized mucin deposition in the stromal connective tissue. While both share similar histological features, they occur in distinct anatomical sites and clinical contexts and have not been directly compared in the literature. **Method:** This study presents a case series of 39 OFM cases diagnosed over 25 years, supplemented by a literature review of previously reported OFM cases, and compares the combined data with published cases of SCFM. The literature-based analysis included 116 OFM cases published in four articles and 138 cases of SCFM published in five articles. Demographic and clinical data were extracted and analyzed, including age, sex, lesion location, size, duration, symptoms, clinical impression, treatment, and recurrence. **Results:** The mean age of OFM patients was 41 years, with a slight female predominance, most commonly affecting the gingiva. SCFM cases were more common in males, with a higher mean age of 52 years and frequent occurrence on the extremities and trunk. Both lesions were predominantly asymptomatic and managed by conservative excision. Due to its rare occurrence and nonspecific clinical presentation, both entities were frequently clinically misdiagnosed. **Conclusions:** In conclusion, this is the first study to directly compare OFM with SCFM and represents the largest series of OFM reported to date. The study provides new comparative insights into SCFM and OFM, highlighting differences in age, gender, lesion site, size, and symptomatology. SCFM predominantly affects older males on the extremities, whereas OFM occurs in younger females, mainly in the gingiva, with larger, sometimes symptomatic lesions, and with a very low recurrence rate.

## 1. Introduction

Mucinoses represent a group of disorders characterized by the pathological accumulation of mucin in the connective tissues of the skin or oral mucosa. These lesions can be localized or part of a systemic disorder [1]. Among the localized variants are solitary cutaneous focal mucinosis (SCFM) and oral focal mucinosis (OFM). Both entities are benign and histologically similar but differ in anatomical location, patient demographics and clinical presentation [2].

SCFM was first described by Johnson and Helwig in 1966 as a solitary, asymptomatic, skin-coloured papule or nodule [3]. It is characterized by the deposition of mucin in the upper dermis, and most commonly, the extremities and trunk of middle-aged male patients are affected [1]. While the etiology is undetermined, local trauma has been suggested as a possible trigger [1]. SCFM is typically managed with simple surgical excision [1].

OFM, introduced by Tomich et al. in 1974, is regarded as the oral counterpart of SCFM [4]. It is an uncommon tumour-like mass similarly characterized as SCFM by localized myxoid changes in the connective tissue. It is of unknown etiology and thought to occur as a result of over production of hyaluronic acid by fibroblasts [4]. OFM commonly presents on the gingiva of middle-aged female patients as an asymptomatic nodule or swelling [2]. Being a rare, reactive lesion, it is often misdiagnosed clinically as other reactive lesions such as epulides. Like SCFM, the treatment of choice is conservative surgical excision [5] and OFM, once excised rarely recurs.

Despite similar clinical and histological features, OFM and SCFM are rarely compared in the literature. This study aimed to present the largest OFM series to date and compare its clinicopathological features with SCFM to clarify distinctions and enhance diagnostic accuracy.

## 2. Materials and Methods

This study was designed as a retrospective case series and literature review. It included 39 previously unpublished cases of OFM retrieved from the archives of the Department of Oral Pathology, Faculty of Dental Sciences, University of Peradeniya, supplemented with an additional 116 published OFM cases identified from the literature. For SCFM, all available cases (138) reported in the literature were included for comparison. Studies were included if they met the following inclusion criteria. Population: human subjects diagnosed with OFM or SCFM. Diagnosis: confirmed through histopathological examination. Location: lesions located in the oral cavity (OFM) or skin (SCFM). Study design: case reports, case series, systematic reviews of case series. Language: published in English. Studies were excluded if they were not peer-reviewed (e.g., letters to the editor, conference abstracts), focused on animal models or in vitro studies, or did not provide sufficient clinical or histopathological data.

A comprehensive synthesis of the literature was conducted, focusing on clinical features: age, gender, lesion size, location, and symptoms; histopathological findings: myxoid degeneration and cellular characteristics; diagnostic methods: techniques used for diagnosis, including staining and imaging; treatment approaches: surgical excision, recurrence rates, and follow-up outcomes; and comparative analysis: differences and similarities between OFM and SCFM.

### 2.1. Case Series

The 39 cases of OFM retrieved from the archives of the Department of Oral Pathology, were diagnosed over a period of 25 years from 2000 to 2025. Data on patient demographics, including age, sex and lesion site, size, duration, morphology, number of lesions, symptoms, histopathology, management and recurrence, were obtained from clinical records and pathology databases and evaluated. Histopathological diagnoses of OFM were confirmed by one of the authors (PRJ) prior to its inclusion.

### 2.2. Study Selection and Data Extraction from the Literature

#### 2.2.1. OMF

A literature review was conducted to compare the clinical and pathological spectrum of the present series with previously reported cases of OFM in the literature. Electronic searches without publication date restriction were undertaken in May 2025 in the following databases: PubMed/MEDLINE and Google Scholar. Search strategy included terms [“oral focal mucinosis” [Title/Abstract] OR “solitary oral focal mucinosis” [Title/Abstract] OR “OFM” [Title/Abstract] OR “oral focal myxoid lesion” [Title/Abstract] restricted to the years 2000–2025. A total of 42 publications were found in electronic searches for OFM (Figure 1). After elimination of duplicates and application of inclusion and exclusion criteria, a total of 37 full texts were assessed for eligibility. Records included in the systematic review article were excluded (n = 29), along with 4 other articles for full-text unavailability (n = 3) and language restrictions (n = 1). A total of 4 articles were included in the literature review. This included the OFM systematic review article published in 2024 and 3 other case reports from Iran, India, and Japan. Titles and abstracts of all references retrieved through the electronic searches were read. If the title and abstract met the eligibility criteria, the article was included for full-text reading.

For each included study, the following data were extracted on a standard form: publication details, patient’s sex, age, symptoms, anatomical location of the lesion, size of the lesion, duration, clinical presentation, diagnostic hypothesis, management, and recurrence Figure 1 [5,6,7,8]. Unavailable data were defined as not available (NA).

#### 2.2.2. SCFM

Electronic searches were undertaken in May 2025 in the following databases: PubMed/MEDLINE and Google Scholar. The following search strategy was used: “cutaneous focal mucinosis” [Title/Abstract] OR “solitary cutaneous mucinosis” [Title/Abstract] OR “solitary focal cutaneous mucinosis” [Title/Abstract] OR “SFCM” [Title/Abstract] restricted to the years 2000–2025.

A total of 38 articles were retrieved (Figure 2). Articles describing case reports or case series of SCFM published in English with enough clinical and histopathological information to confirm the diagnosis were included. Articles on oral focal mucinosis, multiple cutaneous mucinosis and unrelated articles were excluded. Five articles included in the review article were also excluded. Data was extracted from 5 articles Figure 2 [1,9,10,11,12].

### 2.3. Statistical Analysis

Descriptive and quantitative data analyses were performed using the Statistical Package for the Social Sciences for Windows, version 20.0 (SPSS, Inc., Chicago, IL, USA). The normality of data was assessed using the Shapiro–Wilk test. As the data were normally distributed, comparisons between the case series of OFM and literature-based OFM with respect to mean age at diagnosis, evolution times of the lesions, and lesion size were performed using the independent-samples *t*-test, while categorical variables (sex distribution and anatomical locations) were compared using the chi-square test. A significance threshold of *p* < 0.05 was applied. Corrections for multiple testing were considered unnecessary as the comparisons were limited.

## 3. Results

The clinical data of the 39 OFM cases of the present study are summarized in Table 1.

The present series (Table 1) comprised 20 females (51.28%) and 19 males (48.71%), with a 1.05:1 female-to-male ratio. The mean age of patients was 43.86 ± 15.80 years (range: 14–70 years). The most common anatomical location was the gingiva (22 cases, 56.41%), followed by the alveolar ridge mucosa (8 cases, 20.51%) and palate (7 cases, 17.95%). Most cases were asymptomatic (36 cases, 92.31%) although a bleeding tendency had been reported in some cases (3 cases, 7.69%). The lesion size averaged at 2.15 ± 1.36 cm. The average duration of lesions prior to diagnosis was 21.25 ± 27.75 months. Clinically, most lesions were diagnosed as reactive lesions (74.19%). The comparison of gingival lesions between maxilla and mandible showed no statistically significant difference (*p* = 0.79). Statistical analysis revealed significant differences in sex distribution and lesion size between the present series and the literature (*p* = 0.027 and *p* = 0.009). The literature demonstrated a marked female predominance, with 70.80% of cases reported in females and a female-to-male ratio of 2.42:1. Comparison with the literature revealed no significant differences in age distribution (*p* = 0.235), anatomical site (*p* = 0.74), and duration (*p* = 0.59). A single case of recurrence was observed, aligning with previously reported findings in the literature. The lesion recurred 7 months after complete surgical excision (Table 2).

### 3.1. Pathological Features

On macroscopic examination, the lesions presented as whitish mucosal nodules. Microscopically, a mucosal nodule covered by stratified squamous epithelium was observed. The corium contained loosely arranged fibrous tissue in a myxoid stroma. A total of 29.41% of cases (10 cases) showed an inflammatory infiltrate (Figure 3a,b). Four cases (10.2%) showed an ulcerated epithelium. Out of the 39 cases of OFM, 17 had an Alcian blue stain performed to show the mucinous material within the stroma (Figure 4a–d). All 17 cases (100%) were positive for the alcian blue stain.

### 3.2. Comparison with Solitary Cutaneous Focal Mucinosis

Comparison of the present OFM series with the literature revealed no significant differences in age distribution, lesion size, anatomical site, and duration. A statistically significant difference in the sex distribution between the present series and the literature was detected (*p* = 0.027). However, both revealed a female predominance. Therefore, the data of the present series was incorporated to the literature review for the comparative analysis with SCFM.

The following data were extracted from the selected SCFM studies in the literature: patient’s sex, mean age, age range, duration of lesion, number of lesions, size of lesion, symptoms, morphology, site, treatment, and recurrence [1,9,10,11,12]. Table 3 compares the demographic features and clinical characteristics of the two entities.

## 4. Discussion

Cutaneous focal mucinosis is a dermal degenerative primary mucinosis. The term was first introduced by Johnson and Helwig in 1966 to describe an asymptomatic solitary lesion on the skin. In 2016, Kuo et al. suggested that this skin lesion be referred to as solitary cutaneous focal mucinosis [1]. OFM is described as the oral counterpart of SCFM. OFM is a rare, benign soft tissue lesion of unknown etiology [13]. It was first described in 1974 by Tomich et al. [4]. To the best of our knowledge, the present study represents the largest series of OFM cases reported in the literature. Furthermore, it is the first clinicopathological comparison of OFM with SCFM.

The pathogenesis of SCFM remains to be determined. It is considered as a benign local reactive mucinosis. It has typically not been associated with systemic diseases. Localized trauma has been hypothesized as an etiologic factor in the development of SCFM [1]. The pathogenesis of OFM is undetermined. Studies have suggested that it is caused by the overproduction of hyaluronic acid by fibroblasts, which causes localized myxomatous changes. However, the reason for this overproduction is unknown. Some authors suggest that local trauma is the predisposing factor, but it remains controversial [4,13]. The present series did not reveal any apparent involvement of masticatory trauma or local irritation. Immunohistochemical studies indicate that these lesions are typically negative for SMA, CD34, and CD68, supporting their non-neoplastic, reactive nature and distinguishing them from other myxoid soft tissue lesions (Table 4).

The observed difference in sex distribution between our case series of OFM and the literature may reflect sampling variability, population-specific factors, and referral or reporting patterns [5,6,7,8]. Smaller sample sizes in previous studies and regional differences could have underrepresented one sex, contributing to the discrepancy with our findings. These variations highlight the need for larger, multi-centre studies to clarify possible epidemiological trends.

While SCFM is more prevalent in males with a male/female ratio of 1.46:1, OFM is more prevalent in females with a male/female ratio of 0.50:1. The mean age of patients with SCFM was 52 years compared to 41 years of OFM patients. All SCFM lesions were asymptomatic as opposed to only 89.76% of OFM lesions. The size of SCFM lesions ranged from 2 to 20 mm with a variable duration, ranging from months to years. Conversely, the OFM lesions ranged in size from 1 to 100 mm, with an average duration of 18.58 months. SCFM commonly affects the skin of extremities (55.07%), the trunk (31.88%), and the head and neck (13.04%). The most common intraoral sites of OFM are the gingiva (52.94%) and palate (16.99%), followed by the alveolar ridge mucosa (13.73%). Occurrence in the lip (1.96%) and the floor of the mouth (0.65%) are rare, with only a few published cases in the literature [5,6,7,8]. Clinically, the SCFM lesion appears as an asymptomatic, solitary nodule or papule on the skin. It is typically flesh-coloured but may also appear white or red [1]. Based on above findings this study provides new comparative insights into SCFM and OFM. While SCFM mainly affects older adults and the extremities with a slight male predominance, OFM occurs in younger individuals with a predilection for gingiva and other oral sites with a female predominance. OFM lesions were larger and more likely to be symptomatic than SCFM, and a small recurrence rate was noted for OFM but not SCFM. These findings highlight clear differences in clinical presentation, lesion size, and anatomical distribution, offering a more detailed understanding than previously available [1,5,6,7,8,9,10,11,12].

The most commonly considered clinical diagnoses of the SCFM lesion were basal cell carcinoma (31%) and nevus (26%). Other differential diagnoses included cyst (12%), seborrheic keratosis (8.5%), and fibroma (7%) [1]. Interestingly, SCFM had never been included in the diagnoses by the clinicians. This suggests the unfamiliarity of dermatologists with this lesion, possibly because it is a very rare condition [3]. Recognizing SCFM is crucial to prevent misdiagnosis as common lesions like molluscum, warts, milia, and cysts [14]. Therefore, it is important to draw attention to include solitary CFM in clinical differential diagnosis of polypoid skin lesions [3].

However, the clinical features of OFM are not specific [2]. Most cases (89.76%) present as asymptomatic solitary nodules or lumps with a fibrous or hyperplastic appearance. The majority of cases (122 cases, 92.42%) had been diagnosed clinically as reactive lesions, such as fibrous epulis, fibroepithelial polyp, pyogenic granuloma, peripheral giant cell granuloma, etc. Furthermore, none of the clinical diagnoses of the present series or those reported in the literature included OFM.

SCFM and OFM share similar histopathological features. A well-circumscribed region of myxoid connective tissue can be observed in both with delicate collagen fibres and scattered spindle shaped fibroblasts. The principal pathological feature of SCFM is the deposition of mucin in the upper dermis. This can extend into the deeper dermis and rarely into the superficial subcutaneous fat [1]. It appears well-localized but unencapsulated. The mucinous change appears lightly basophilically stained on hematoxylin-and-eosin-stained sections (Figure 3). Special stains, such as Alcian blue, have been used to confirm the mucinous material as hyaluronic acid [1]. Fibroblasts are scattered within the mucinous stroma along with reduced amounts of collagen fibres, elastic fibres, and reticulum fibres. The overlying epidermis may be normal, atrophic, or hyperplastic [1]. Table 4 summarizes the diagnostic criteria of OFM and SCFM.

Histopathologically, OFM must be distinguished from odontogenic myxoma, myxoid solitary fibrous tumour, myxoid neurofibroma, and fibrous hyperplasia with myxoid degeneration [2]. Odontogenic myxoma differs by its odontogenic origin, intraosseous location, and aggressive behaviour, necessitating radiological exclusion before confirming OFM. Although OFM lacks specific immunohistochemical markers, IHC is useful for excluding other myxoid lesions—S-100 helps rule out neural tumours, while CD34, STAT6, and BCL2 assist in excluding myxoid solitary fibrous tumour [14,15].

The main histopathological differential diagnoses of SCFM include intramuscular myxoma, superficial angiomyxoma, nodular fasciitis, myxoid dermatofibroma, and dermatofibrosarcoma (DFSP) with myxoid change [1,14,15]. Superficial angiomyxoma shows prominent vasculature, neutrophil-rich stroma, a lobular growth pattern, and CD34-positive spindle cells; multiple lesions require exclusion of Carney complex. Nodular fasciitis is SMA-positive with fascicular spindle cells, whereas SCFM is SMA-negative. Myxoid dermatofibroma is CD68-positive, and DFSP shows CD34-positive spindle cells, both features absent in SCFM. Other mucin-rich dermal lesions that may require exclusion include lupus erythematosus (tumid variant), mucinous nevus, mucinous fibroma, myxoid cyst, and self-healing juvenile cutaneous mucinosis [14,15].

Dermoscopic observations of the SCFM lesion revealed a non-specific homogenous whitish pattern possibly due to the birefringent properties of mucin or reduced number of melanocytes in the epidermis above the dermal mucin, or both [1,16]. Electron microscopy confirmed the presence of stellate fibroblasts scattered in the mucinous stroma along with reduced number of collagen, elastic and reticulum fibres [1]. It is important to emphasize the solitary nature of the lesion of both SCFM and OFM. Thus, avoiding the erroneous use of the terms SCFM and OFM to describe multiple lesions associated with mucinosis-related systemic diseases such as Hashimoto thyroiditis, Birt–Hogg–Dubé syndrome, and Graves’ disease [2]. This differentiation is essential, as the management strategies differ. It has been confirmed that SCFM and OFM patients do not present with any mucinosis-related systemic conditions [2,3]. Therefore, additional laboratory investigations for patient evaluation are not required. Simple surgical excision is the treatment of choice [2,16,17]. In contrast, the presence of multiple lesions should prompt evaluation for an underlying systemic condition [3]. In the present series, all lesions were confined to the oral cavity, and systemic involvement was excluded. Recurrence of the SCFM lesion has not been reported [1]. Only two recurrences of OFM cases have been reported in the literature. The present series also highlighted one recurrence.

The comparative analysis between OFM and SCFM improves diagnostic accuracy by clarifying subtle clinicopathological differences between these two morphologically overlapping entities. By highlighting distinctions in demographics, site, clinical presentation, and histopathological patterns, this study provides practical diagnostic criteria that help prevent misdiagnosis.

## 5. Conclusions

In conclusion, the present study reports the largest series of OFM cases to date, provides the first direct clinicopathological comparison between OFM and SCFM, identifies distinct demographic and site-related patterns (SCFM in older males on extremities; OFM in younger females on gingiva), and emphasizes practical implications for awareness, diagnostic accuracy, and clinical management.


## Figures and Tables

**Figure 1 dermatopathology-12-00038-f001:**
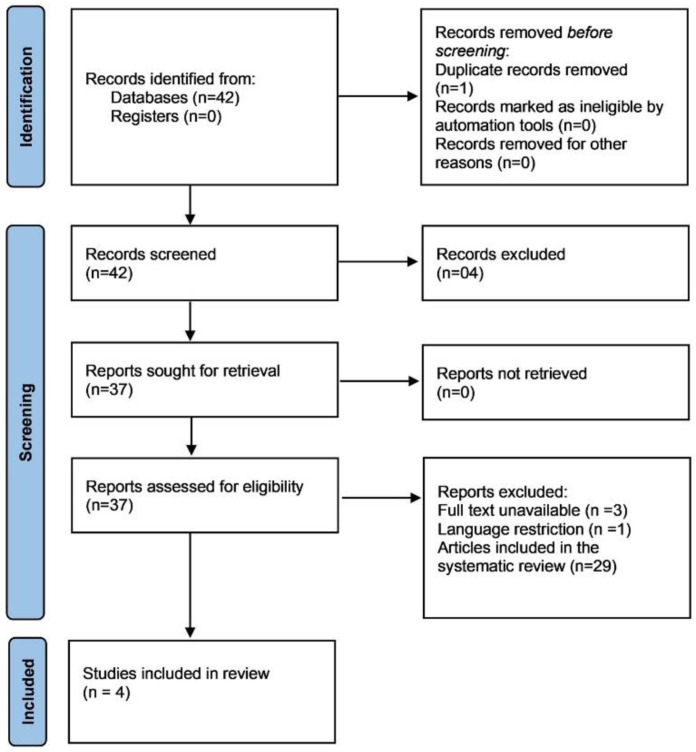
Flowchart illustrating the selection process of studies on OFM.

**Figure 2 dermatopathology-12-00038-f002:**
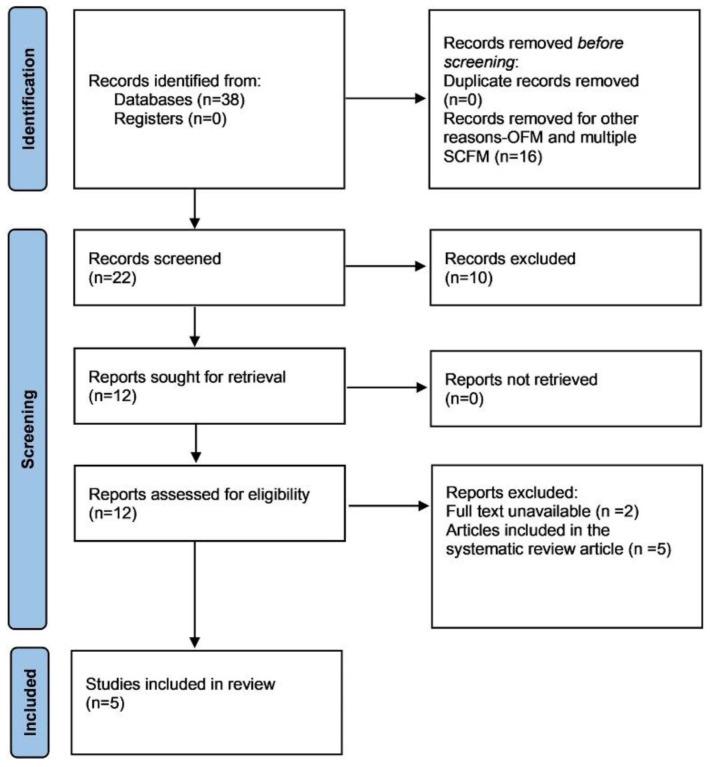
Flowchart illustrating the selection process of studies on SCFM.

**Figure 3 dermatopathology-12-00038-f003:**
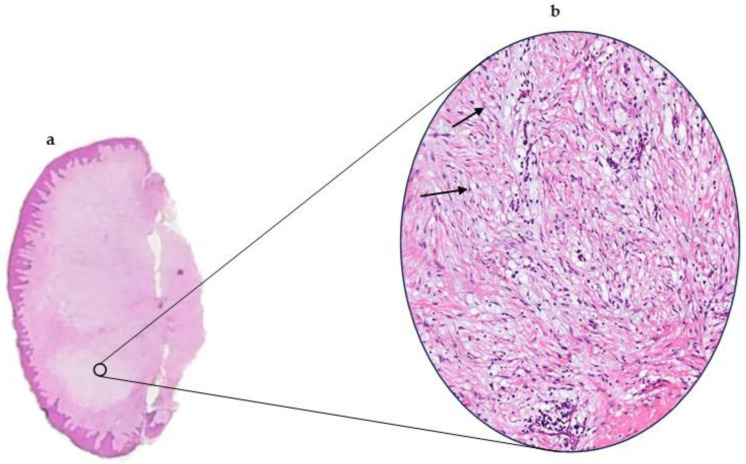
(**a**). OFM presenting as a mucosal nodule covered by parakeratinized stratified squamous epithelium. Note: lightly stained areas that contain the mucoid material (H&E x4). (**b**)**.** High-power photomicrograph with hematoxylin and eosin stain, revealing stellate fibroblasts. Note the fibroblasts indicated by arrows, showing delicate fibrillar processes extending from the fibroblast cytoplasm.

**Figure 4 dermatopathology-12-00038-f004:**
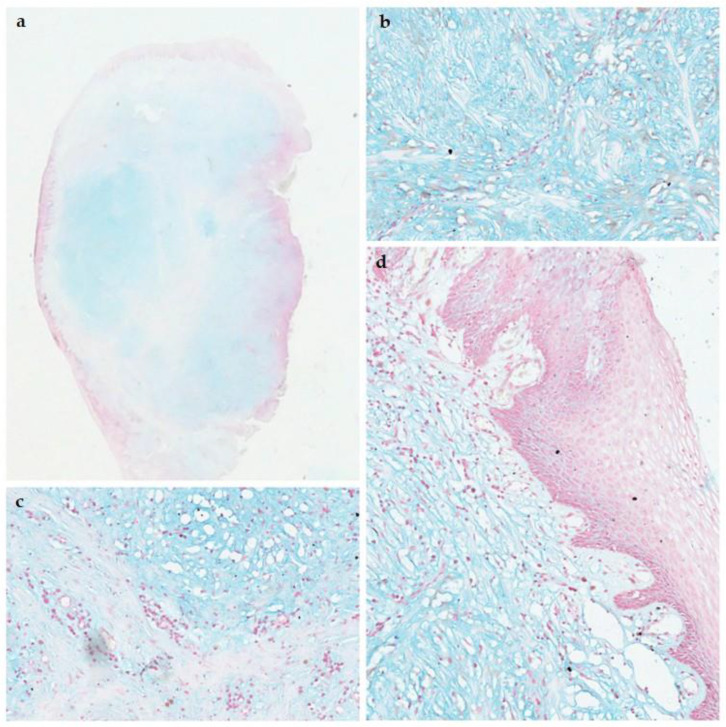
(**a**). Photomicrograph showing Alcian blue positive mucoid material arranged into nodules within the lesion (x4), (**b**–**d**). High-power views of the Alcian blue positive mucoid material. (**c**) Mucoid material separated by dense collagen fibres is shown (Alcian blue stain).

**Table 1 dermatopathology-12-00038-t001:** Summary of demographic and clinical characteristics in the present series of oral focal mucinosis.

Case	Sex/Age	Location	Duration (Months)	Size (cm)	Symptoms	Clinical impression	Treatment
1	M/31	Palate	12	4	Asymptomatic	Pleomorphic adenoma	Ex
2	F/70	Gingiva	NA	NA	Asymptomatic	NA	Ex
3	F/33	Gingiva	12	2	Asymptomatic	Suspicious lesion	Ex
4	M/14	Gingiva	3	2	Asymptomatic	Suspicious lesion	Ex
5	F/44	Gingiva	3	1	Bleeding tendency	Fibrous epulis	Ex
6	M/54	Palate	NA	NA	Asymptomatic	Traumatic lesion	In
7	F/NA	Gingiva	12	1.5	Bleeding on brushing	Epulis	Ex
8	F/38	Gingiva	2	3	Asymptomatic	Fibroepithelial polyp	In
9	F/21	Gingiva	NA	1	Asymptomatic	Osteoma	Ex
10	F/57	Alveolar ridge mucosa	4	2	Asymptomatic	Fibrous epulis	Ex
11	M/33	Palate	36	2	Asymptomatic	Fibroepithelial polyp	Ex
12	F/59	Gingiva	3	2	Asymptomatic	NA	Ex
13	M/48	Buccal sulcus	2.5	5	Asymptomatic	Giant cell tumour	Ex
14	F/59	Gingiva	4	1	Asymptomatic	Fibrous epulis	In
15	F/49	Alveolar ridge mucosa	NA	NA	Asymptomatic	NA	Ex
16	M/48	Palate	12	3	Asymptomatic	NA	Ex
17	F/60	Alveolar ridge mucosa	2.5	3	Asymptomatic	Fibroepithelial polyp	Ex
18	F/28	Palate	NA	0.3	Asymptomatic	Fibroepithelial polyp	Ex
19	M/58	Alveolar ridge mucosa	3	2.5	Asymptomatic	Recurrent - Oral focal mucinosis	Ex
20	M/58	Gingiva	NA	3	Asymptomatic	Gingival epulis	Ex
21	M/NA	Palate	36	NA	Asymptomatic	Papilloma	Ex
22	M/48	Gingiva	NA	1	Easily bleeding	Granuloma	Ex
23	F/14	Gingiva	0.5	1	Asymptomatic	Pyogenic granuloma	Ex
24	M/44	Alveolar ridge mucosa	6	2	Asymptomatic	Pyogenic granuloma	Ex
25	F/48	Gingiva	NA	NA	Asymptomatic	Giant cell granuloma	Ex
26	M/65	Gingiva	12	3	Asymptomatic	NA	NA
27	F/36	Gingiva	7	4	Asymptomatic	Fibrous epulis	Ex
28	M/56	Alveolar ridge mucosa	12	0.8	Asymptomatic	Fibrous epulis	Ex
29	F/19	Gingiva	24	0.5	Asymptomatic	Fibroepithelial polyp	Ex
30	F/40	Alveolar ridge mucosa	3.5	0.5	Asymptomatic	Fibroepithelial polyp	Ex
31	F/62	Gingiva	2	NA	Asymptomatic	Odontogenic fibroma	Ex
32	M/36	Gingiva	144	3	Asymptomatic	Fibroepithelial polyp	Ex
33	M/64	Retromolar region	NA	NA	Asymptomatic	NA	In
34	M/21	Gingiva	8	NA	Asymptomatic	Fibroepithelial polyp	NA
35	F/51	Palate	1	1	Asymptomatic	Fibroepithelial polyp	Ex
36	M/15	Gingiva	NA	2	Asymptomatic	Non-specific localized gingival hypertrophy	NA
37	M/NA	Gingiva	42	1.5	Asymptomatic	Gingival polyp	NA
38	F/53	Gingiva	0.5	6	Asymptomatic	Malignant lesion	NA
39	M/45	Alveolar ridge mucosa	NA	NA	Asymptomatic	NA	Ex

Abbreviations: Ex, excision; F, female; In, incision biopsy; M, male; NA, not available. Some data for OFM, particularly lesion size, clinical impression, and duration, were unavailable; however, these omissions are unlikely to have significantly influenced the overall clinicopathological comparison or the main conclusions, as more than 75% of the data were available.

**Table 2 dermatopathology-12-00038-t002:** Summary of demographic and clinical features of oral focal mucinosis: present series vs. literature review [1,5,6,7,8,9,10,11,12].

Variable	Present Series	Literature Review
Sex	n = 39	n = 116
Female	20 (51.28%)	83 (71.55%)
Male	19 (48.71%)	33 (28.45%)
Ratio	1.05: 1	2.52: 1
Age (years)	n = 36	n = 113
Mean (SD)	43.86 (15.80)	40.74 (17.79)
Range	14–70	2–88
Not informed	3	
Anatomical location	n = 39	n = 114
Gingiva	22 (56.41%)	59 (51.75%)
Palate	7 (17.95%)	19 (16.67%)
Alveolar ridge mucosa	8 (20.51%)	13 (11.40%)
Buccal mucosa	1 (2.56%)	4 (3.51%)
Retromolar region	1 (2.56%)	5 (4.39%)
Tongue		10 (8.77%)
Lip		3 (2.63%)
Floor of the mouth		1 (0.88%)
Recurrence	n = 39	n = 69
Yes	1 (2.56%)	1 (1.45%)
No	38 (97.44%)	68 (98.55%)
Lesion size (cm)	n = 30	n = 86
Mean (SD)	2.15 (1.36)	1.37 (1.41)
Range	0.3–6	0.1–10
Duration of the lesion (months)	n = 28	n = 72
Mean (SD)	21.25 (27.75)	17.55 (23.83)
Range	0.5–144	1–120
Management	n = 34	n = 104
Surgical excision	30 (88.24%)	98 (94.23%)
Incisional biopsy	4 (11.76%)	6 (5.77%)
Clinical hypothesis	n = 31	n = 101
Reactive lesions	23 (74.19%)	99 (98.02%)
Benign tumours	5 (16.13%)	
Malignant lesions	3 (9.68%)	2 (1.98%)

**Table 3 dermatopathology-12-00038-t003:** Comparison of demographic and clinical features of solitary cutaneous focal mucinosis and oral focal mucinosis.

	Solitary Cutaneous Focal Mucinosis	Oral Focal Mucinosis
Cases	138	155
Men/women	1: 0.6	0.5: 1
Mean age (years)	52	41
Age range (years)	6–86	2–88
Duration of lesion	Months-years	18.58 months
Number of lesions	1	1
Symptoms	Asymptomatic = 138 (100%)	Asymptomatic = 114 (89.76%) Symptomatic = 13 (10.24%)
Size (mm)	2–20	1–100
Morphology	Nodule Papule Mass	Swelling Nodule Lump
Site	Extremities = 76 (55.07%) Trunk = 44 (31.88%) Head and neck = 18 (13.04%)	Gingiva = 81 (52.94%) Palate = 26 (16.99%) Alveolar ridge mucosa = 21 (13.73%) Buccal mucosa = 5 (3.27%) Retromolar region = 6 (3.92%) Tongue = 10 (6.54%) Lip = 3 (1.96%) Floor of the mouth = 1 (0.65%)
Treatment	Excision = 26 Other (biopsy/curettage/electrodessication and curettage) = 3	Excision = 128 (89.51%) Other (incisional biopsy) = 10 (6.99%)
Recurrence	0%	1.85%

**Table 4 dermatopathology-12-00038-t004:** Diagnostic criteria of solitary cutaneous focal mucinosis and oral focal mucinosis.

Feature	SCFM	OFM
Architecture	Well-circumscribed, non-encapsulated dermal lesion.	Well-circumscribed, non-encapsulated lesion in the lamina propria.
Stroma	Mucin-rich (myxoid) dermis.	Abundant myxoid (mucin-rich) connective tissue.
Cells	Scattered spindle cells; no fascicular or storiform pattern.	Scattered spindle-shaped fibroblasts; lacks fascicular or storiform arrangement.
Vascularity	Usually not prominent; differs from superficial angiomyxoma.	Generally sparse, without prominent vasculature.
Inflammation	Minimal or absent inflammatory infiltrate; neutrophils rare.	Minimal or absent inflammatory infiltrate.
Special stains	Positive for Alcian Blue.	Positive for Alcian Blue (acidic mucopolysaccharides).
Immunohistochemistry	SMA-negative; CD34-negative; CD68-negative.	Typically SMA-negative; CD34-negative; CD68-negative.

## Data Availability

The dataset of this study is available upon reasonable request from the corresponding authors. Due to ethical restrictions, the data are not publicly accessible.

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
