# Peer review of "Comparative Clinicopathological Analysis of Oral Focal Mucinosis and Solitary Cutaneous Focal Mucinosis: A Case Series and Literature-Based Analysis"

_dermatopathology, 2025, doi:10.3390/dermatopathology12040038_

Round 1

Reviewer 1 Report

Comments and Suggestions for Authors

Are the vessels always present in the histology, or are not important for the diagnosis? becaue OFM is regarded a countepart of SCFM, I suggest a prompt evauation of systemic conditions of a solitary lesion of unknown nature. Wich the first diagnosis when you see The SCFM for the first time?

Author Response

Reviewer 1

Comments and Suggestions for Authors

Comment 1: Are the vessels always present in the histology, or are not important for the diagnosis?

Response 1: Presence of vessels is considered as a non-specific finding and not important for the diagnosis.

Comment 2: because OFM is regarded a countepart of SCFM, I suggest a prompt evauation of systemic conditions of a solitary lesion of unknown nature.

Response 2: Solitary lesions of OFM, does not warrant evaluation of systemic conditions, purely because it is multiple lesions that are associated with systemic disease.

Comment 3: Which the first diagnosis when you see The SCFM for the first time?

Response 3: SCFM typically presents as an asymptomatic dome shaped nodule or papule, with the colour ranging from flesh coloured to-red-white. Given the variable morphological presentation, SCFM is rarely clinically diagnosed. Clinical differential diagnoses may range from adnexal tumours to soft tissue tumours.

Reviewer 2 Report

Comments and Suggestions for Authors

I would like to thank you for the opportunity to review this interesting and original manuscript, which addresses the comparison between oral focal mucinosis and solitary cutaneous focal mucinosis. The study provides valuable clinical and pathological insights and presents the largest case series of OFM to date. However, I believe that several important aspects require clarification and further improvement before the paper can be considered for publication. For this reason, I recommend major revisions.

Detailed comments and suggestions aimed at strengthening the clarity, methodological rigor, and overall presentation of the work are provided in the attached annotated PDF.

Comments on the Quality of English Language

The quality of the English language is overall acceptable and the manuscript is understandable. Nevertheless, several sentences are lengthy and occasionally repetitive, which makes the reading less fluent. Some passages, particularly in the Introduction and Discussion, would benefit from stylistic refinement to improve clarity and conciseness. A careful language editing by a native or professional scientific editor is recommended to enhance readability and ensure consistency throughout the manuscript.

Author Response

Reviewer 2

I would like to thank you for the opportunity to review this interesting and original manuscript, which addresses the comparison between oral focal mucinosis and solitary cutaneous focal mucinosis. The study provides valuable clinical and pathological insights and presents the largest case series of OFM to date. However, I believe that several important aspects require clarification and further improvement before the paper can be considered for publication. For this reason, I recommend major revisions.

Detailed comments and suggestions aimed at strengthening the clarity, methodological rigor, and overall presentation of the work are provided in the attached annotated PDF.

peer-review-50165913.v1.pdf (see below)

I would like to thank the Editors of Dermatopathology for the opportunity to review this interesting manuscript, and I congratulate the Authors for addressing the clinicopathological comparison between oral focal mucinosis (OFM) and solitary cutaneous focal mucinosis (SCFM). The topic is original and potentially valuable for both dermatologists and oral pathologists. However, several issues need clarification and improvement before the manuscript can be considered for publication.For this reason, I recommend major revisions.

Comment 1: Title and Abstract

  • The title is clear and descriptive, but could be shortened for conciseness (e.g., avoid

“literature-based analysis” and specify “comparative clinicopathological study”).

Response 1: Thank you for the suggestion. Title was modified as follows, however, as major component of the study is based on published literature we would like to keep that component in the title

Comparative Clinicopathological Analysis of Oral Focal Mucinosis and Solitary Cutaneous Focal Mucinosis: A Case Series and Literature-Based Analysis

Comment 2: The abstract provides a good overview, but the methodology is described too briefly. For example, the literature search strategy for OFM and SCFM is not detailed. Consider specifying inclusion/exclusion criteria and number of final articles included.

Response 2: In order to follow the reviewers suggestion, following sentence was added to the abstract to improve the methodology.

The literature based analysis included 116 OFM cases published in 4 articles and 138 cases of SCFM published in 5 articles

Search strategy and inclusion exclusion criteria were specified in the methodology section to keep within the word count of the abstract

Comment 3: Line 69-72-75 are used to expand the literature search strategy and details are also given in figure 1 and 2.

Response 3: Articles describing case reports or case series of oral focal mucinosis published in English with enough clinical and histopathological information to confirm the diagnosis were included. Duplicate and irrelevant reports were excluded

Comment 4: The conclusion of the abstract should be more specific: rather than generic recommendations, highlight the novel contribution of your comparison (e.g., that OFM and SCFM share histological features but differ in sex and age distribution).

Response 4: Thank you for the suggestion-following sentence was added to the conclusion

 In conclusion, this study provides new comparative insights into SCFM and OFM, highlighting differences in age, gender, lesion site, size and symptomatology. SCFM predominantly affects older males on the extremities, whereas OFM occurs in younger females, mainly in the gingiva, with larger, sometimes symptomatic lesions and with a very low recurrence rate.

Comment 5: Introduction

The introduction provides useful background but is slightly repetitive. For example, the

description of OFM and SCFM histological features overlaps with the Discussion.

Response 5: We agree that there is some overlap in the description of histological features between the Introduction and Discussion. However, we believe it is important to briefly outline the key histopathological features of both OFM and SCFM in the Introduction to set the stage for the comparative clinicopathological analysis. This provides readers with the necessary background for understanding that both lesions share similar histopathological features, while the Discussion elaborates on additional features in more detail. Therefore, we would like to keep the histopathological description in both places

Comment 6: The rationale and objectives of the study should be stated more clearly at the end. At present, it is not entirely clear whether the main goal is to describe the largest OFM series, to compare OFM and SCFM, or both.

Thank you for this comment, accordingly the objective was rephrased as follows

Response 6: The objective of this study was to present the largest series of OFM to date and to conduct a comparative clinicopathological analysis of OFM and SCFM, thereby clarifying their similarities and differences in order to improve awareness and diagnostic accuracy.

Comment 7: Materials and Methods

The case series section is well explained, but the study design (retrospective case series +

literature review) should be stated explicitly at the beginning.

Following paragraph was used to address the reviewers comment

Response 7: This study was designed as a retrospective case series and literature review. It included 39 previously unpublished cases of OFM retrieved from the archives of the Department of Oral Pathology, Faculty of Dental Sciences, University of Peradeniya supplemented with additional 116 published OFM cases identified from the literature. For Solitary Cutaneous Focal Mucinosis (SCFM), all available cases (138) reported in the literature were included for comparison.

Comment 8:  The literature review methodology is insufficiently detailed. Please clarify:

Following sentence was added. Literature review methodology is illustrated in fig 1 and 2. Therefore, additional sentences were not added. However, if the reviewer still feels that methodology is insufficient with fig1 and 2, further explanations can be added during the 2nd round of review

Response 8: Search strategy included terms [“oral focal mucinosis” [Title/Abstract] OR “solitary oral focal mucinosis” [Title/Abstract] restricted to the years 2000-2025. Duplicate and irrelevant reports were excluded.

Comment 9: Were PRISMA guidelines followed?

Response 9: Yes. Figure 1 and 2 were generated following PRISMA  guideline, with modifications as our study is not a systematic review

Comment 10: Why were systematic review articles excluded but then cited later?

Response 10: As per the figure 1, articles included in the systematic review were excluded. Not the systematic review

Comment 11: Were two independent reviewers involved in study selection?

Response 11: Yes. SMSN Wickramasinghe and PR Jayasooriya (1st and 2nd authors).

Comment 12: The inclusion of Google Scholar as a database should be justified, as it may reduce reproducibility.

Response 12: Due to limited access to subscription-based databases, Google Scholar was used for the literature search. While this may affect reproducibility, it ensured broad coverage of both indexed and non-indexed publications.

Comment 13: Statistical methods are briefly mentioned but not detailed (e.g., which comparisons were tested, what threshold for significance was applied, whether corrections for multiple testing were used).

Response 13: Following was added to the manuscript

As the data were normally distributed, comparisons between the case series of OFM and literature-based OFM with respect to mean age at diagnosis, evolution times of the lesions and lesion size were performed using the independent-samples t-test, while categorical variables (sex distribution and anatomical locations) were compared using the chi-square test. A significance threshold of p < 0.05 was applied. No corrections for multiple testing were performed, as the comparisons were limited.

Comment 14: Results

The clinical table is comprehensive, but many fields are missing (NA). Consider summarizing the most relevant findings in the text to improve readability.

Response 14: We agree with the reviewer’s observation. However, as the findings have already been summarized in Table 2 and in the paragraph immediately preceding it, we feel that repeating them here would lead to redundancy

Comment 15: Figures 1 and 2 (flowcharts) resemble PRISMA diagrams but are incomplete in formatting.

Standardize them to improve clarity.

Response 15: Figures 1 and 2 were adapted from the PRISMA 2020 flow diagram. The “reports sought for retrieval” step was omitted, as the number was identical to reports assessed for eligibility. This article is a case series with a literature-based comparison and is not a systematic review.

  • Comment 16: Pathological features are described clearly, but the inclusion of quantitative data (percentage of cases with Alcian blue positivity, recurrence rate, etc.) would strengthen the findings.

Response 16: The number of cases on which alcian blue stain was performed was added to the text. Recurrences are given in Table 2. Line 204-205

Comment 17: Discussion

The Discussion is extensive and demonstrates good knowledge of the literature. However:

It often reads as a textbook review rather than a critical discussion of your findings.

Please emphasize what is new in your series compared with existing data.

Following sentences highlighting our findings were added

Response 17: Based on above findings this study provides new comparative insights into SCFM and OFM. While SCFM mainly affects older adults and the extremities with a slight male predominance, OFM occurs in younger individuals with a predilection for gingiva and other oral sites with a female predominance. OFM lesions were larger and more likely to be symptomatic than SCFM, and a small recurrence rate was noted for OFM but not SCFM. These findings highlight clear differences in clinical presentation, lesion size, and anatomical distribution, offering a more detailed understanding than previously available.

Comment 18: Some sections (e.g., differential diagnosis of SCFM) are too detailed and digressive, overshadowing your own results. Consider shortening.

Response 18: We shortened the DD as follows

The main histopathological differential diagnoses of SCFM include intramuscular myxoma, superficial angiomyxoma, nodular fasciitis, myxoid dermatofibroma, and Dermatofibrosarcoma (DFSP) with myxoid change [1, 14, 15]. Superficial angiomyxoma shows prominent vasculature, neutrophil-rich stroma, a lobular growth pattern, and CD34-positive spindle cells; multiple lesions require exclusion of Carney complex. Nodular fasciitis is SMA-positive with fascicular spindle cells, whereas SCFM is SMA-negative. Myxoid dermatofibroma is CD68-positive, and DFSP shows CD34-positive spindle cells, both features absent in SCFM. Other mucin-rich dermal lesions that may require exclusion include lupus erythematosus (tumid variant), mucinous nevus, mucinous fibroma, myxoid cyst, and self-healing juvenile cutaneous mucinosis [14,15].

Comment 19: The significance of the sex distribution difference between OFM series and literature should be further interpreted.

Response 19: The observed difference in sex distribution between our case series of OFM and the literature may reflect sampling variability, population-specific factors, and referral or reporting patterns. Smaller sample sizes in previous studies and regional differences could have underrepresented one sex, contributing to the discrepancy with our findings.

Comment 20: Clarify the clinical relevance of comparing OFM and SCFM—does this comparison improve diagnostic accuracy or change management? This paragraph was added to the discussion

Response 20: Comparing OFM and SCFM improves diagnostic accuracy by creating awareness of the clinical and histopathological features of both conditions, helping clinicians and pathologists distinguish between them and manage them appropriately.

Comment 21: Conclusions

The conclusions are too general and simply restate background knowledge. They should be

sharpened to reflect the main contributions of the present study, e.g.:

o reporting the largest OFM case series;

o highlighting the similarities and differences between OFM and SCFM;

o reinforcing the importance of awareness among dermatologists and oral pathologists.

Response 20: Thank you for the suggestion, conclusion was modified as follows

In conclusion, by reporting the largest OFM case series to date, this study provides new comparative insights into OFM and SCFM, highlighting their similarities and differences in age, gender, lesion site, size, and symptomatology. SCFM predominantly affects older males on the extremities, whereas OFM occurs in younger females, mainly in the gingiva, presenting as larger, occasionally symptomatic lesions with a very low recurrence rate. These findings reinforce the importance of awareness among clinicians and pathologists to improve diagnostic accuracy and ensure appropriate patient management

Comment 21: Comments on the Quality of English Language

The quality of the English language is overall acceptable and the manuscript is understandable. Nevertheless, several sentences are lengthy and occasionally repetitive, which makes the reading less fluent. Some passages, particularly in the Introduction and Discussion, would benefit from stylistic refinement to improve clarity and conciseness. A careful language editing by a native or professional scientific editor is recommended to enhance readability and ensure consistency throughout the manuscript.

Response 21: As requested modifications were made to improve the clarity

Comment 22: References

  • The references are appropriate but need careful checking:

o Some case reports (e.g., Cureus references) are cited multiple times unnecessarily.

o Ensure that all references are consistent with the journal’s style (e.g., punctuation, DOI

formatting).

Done

Reviewer 3 Report

Comments and Suggestions for Authors

The authors present a retrospective analysis of their archival material to identify and describe cases of oral focal mucinosis. The manuscript is generally well organised, and the cases are clearly presented. However, in its current form, I cannot recommend the manuscript for publication. Several important issues need to be addressed:

  1. The literature review is insufficient and does not follow established guidelines such as the PRISMA statement for case series and systematic reviews. The authors should update and transparently report their search strategy, inclusion and exclusion criteria, and provide a more comprehensive synthesis of the relevant literature.
  2. At present, the section describing the review of cutaneous lesions is embedded within the statistical analysis. This is confusing. The literature review should be presented separately from the statistical methods applied to the retrospective case series.
  3. The manuscript lacks a clear description of the diagnostic criteria used to confirm oral focal mucinosis. The authors must explain how the diagnosis was established and what steps were taken to exclude important differential diagnoses such as myxoid neurofibroma, odontogenic myxoma, mucocele, or fibrous hyperplasia with myxoid change. Many of the cases included in the series had initial clinical impressions such as fibroepithelial polyp or epulis. This underscores the need to clearly justify the histopathological basis on which they were reclassified as oral focal mucinosis.
  4. Only a small subset of cases underwent Alcian blue staining to demonstrate mucin. This is a major limitation. Alcian blue (or equivalent mucin stains) should be performed in all cases to confirm the diagnosis, especially in a study of this scale. Without this, diagnostic certainty remains questionable.
  5. The statistical analysis section mixes original case data with information extracted from the literature review. This should be clearly separated:
    • Statistical testing should be applied only to the retrospective case series.
    • The literature review findings should be summarized descriptively in a distinct section.
  1. The discussion of pathogenesis is underdeveloped. The authors briefly mention trauma and fibroblast hyaluronic acid overproduction but provide no integration of recent molecular or immunohistochemical insights. This section should be expanded. The clinical implications of recurrence and misdiagnosis are not sufficiently discussed. The recurrence data (including their one recurrent case) need to be contextualized in terms of management and follow-up.
  2. The conclusion currently overstates clinico-pathological correlations. The data are valuable but should be presented more cautiously, acknowledging diagnostic challenges and limitations.
  3. Given that the study is clinically oriented, the manuscript would benefit from a section offering practical diagnostic guidance for clinicians and pathologists. For example, a summary table or diagnostic checklist highlighting key histopathological and clinical features that distinguish oral focal mucinosis from its mimics.

Author Response

Reviewer 3

The authors present a retrospective analysis of their archival material to identify and describe cases of oral focal mucinosis. The manuscript is generally well organised, and the cases are clearly presented. However, in its current form, I cannot recommend the manuscript for publication. Several important issues need to be addressed:

  1. The literature review is insufficient and does not follow established guidelines such as the PRISMA statement for case series and systematic reviews. The authors should update and transparently report their search strategy, inclusion and exclusion criteria, and provide a more comprehensive synthesis of the relevant literature.

Response: Additional information was provided as follow

This study was designed as a retrospective case series and literature review. It included 39 previously unpublished cases of OFM retrieved from the archives of the Department of Oral Pathology, Faculty of Dental Sciences, University of Peradeniya supplemented with additional 116 published OFM cases identified from the literature. For SCFM, all available cases (138) reported in the literature were included for comparison. Studies were included if they met the following criteria (Inclusion criteria): Population: Human subjects diagnosed with OFM or SCFM. Diagnosis: Confirmed through histopathological examination. Location: Lesions located in the oral cavity (OFM) or skin (SCFM). Study Design: Case reports, case series, systematic reviews of case series. Language: Published in English. Studies were excluded if they: Were not peer-reviewed (e.g., letters to the editor, conference abstracts). Focused on animal models or in vitro studies and did not provide sufficient clinical or histopathological data.

 A comprehensive synthesis of the literature was conducted, focusing on: Clinical Features: Age, gender, lesion size, location, and symptoms, Histopathological Findings: Myxoid degeneration, and cellular characteristics, Diagnostic Methods: Techniques used for diagnosis, including staining and imaging, Treatment Approaches: Surgical excision, recurrence rates, and follow-up outcomes, Comparative Analysis: Differences and similarities between OFM and SCFM

  1. At present, the section describing the review of cutaneous lesions is embedded within the statistical analysis. This is confusing. The literature review should be presented separately from the statistical methods applied to the retrospective case series.

Response: Thank you for pointing it out. It was a mistake and corrected

  1. The manuscript lacks a clear description of the diagnostic criteria used to confirm oral focal mucinosis. The authors must explain how the diagnosis was established and what steps were taken to exclude important differential diagnoses such as myxoid neurofibroma, odontogenic myxoma, mucocele, or fibrous hyperplasia with myxoid change. Many of the cases included in the series had initial clinical impressions such as fibroepithelial polyp or epulis. This underscores the need to clearly justify the histopathological basis on which they were reclassified as oral focal mucinosis.

Response: Histopathological criteria for establishing the diagnosis is indicated in both introduction and differential diagnosis have been discussed in the discussion. However, table 4 was added with a summary of diagnostic criteria

Feature

SCFM

OFM

Architecture

Well-circumscribed, non-encapsulated dermal lesion.

Well-circumscribed, non-encapsulated lesion in the lamina propria.

Stroma

Mucin-rich (myxoid) dermis.

Abundant myxoid (mucin-rich) connective tissue.

Cells

Scattered spindle cells; no fascicular or storiform pattern.

 Scattered spindle-shaped fibroblasts; lacks fascicular or storiform arrangement.

Vascularity

Usually not prominent; differs from superficial angiomyxoma.

Generally sparse, without prominent vasculature.

Inflammation

 Minimal or absent inflammatory infiltrate; neutrophils rare.

Minimal or absent inflammatory infiltrate.

Special stains

Positive for Alcian Blue.

Positive for Alcian Blue (acidic mucopolysaccharides).

Immunohistochemistry

 SMA-negative; CD34-negative; CD68-negative.

Typically SMA-negative; CD34-negative; CD68-negative

  1. Only a small subset of cases underwent Alcian blue staining to demonstrate mucin. This is a major limitation. Alcian blue (or equivalent mucin stains) should be performed in all cases to confirm the diagnosis, especially in a study of this scale. Without this, diagnostic certainty remains questionable.

Response: Initially though only a few cases had alcian blue staining, due to the reviewers comment alcian blue staining was performed on additional cases amounting to a total of 17 confirming the diagnosis. It was not possible to perform the stain on all cases due to damaged wax blocks.

  1. The statistical analysis section mixes original case data with information extracted from the literature review. This should be clearly separated:
    • Statistical testing should be applied only to the retrospective case series.
    • The literature review findings should be summarized descriptively in a distinct section.

Response: The purpose of statistically analyzing the case series alongside cases identified in the literature was to determine whether there were significant differences between the two groups. The rationale was that if major differences existed, it would not be justifiable to combine the groups for a single comparison between OFM and SCFM. However, since only two significant differences were observed—female predilection and lesion size—the two datasets were combined for subsequent analysis.

  1. The discussion of pathogenesis is underdeveloped. The authors briefly mention trauma and fibroblast hyaluronic acid overproduction but provide no integration of recent molecular or immunohistochemical insights. This section should be expanded. The clinical implications of recurrence and misdiagnosis are not sufficiently discussed. The recurrence data (including their one recurrent case) need to be contextualized in terms of management and follow-up.

In spite of additional literature search, authors were unable to identify additional factors on pathogenesis. Following sentence was added

Response: Immunohistochemical studies indicate that these lesions are typically negative for SMA, CD34, and CD68, supporting their non-neoplastic, reactive nature and distinguishing them from other myxoid soft tissue lesions

  1. The conclusion currently overstates clinico-pathological correlations. The data are valuable but should be presented more cautiously, acknowledging diagnostic challenges and limitations.

Response: Conclusion was changed as per follows

In conclusion, by reporting the largest OFM case series to date, this study provides new comparative insights into OFM and SCFM, highlighting their similarities and differences in age, gender, lesion site, size, and symptomatology. SCFM predominantly affects older males on the extremities, whereas OFM occurs in younger females, mainly in the gingiva, presenting as larger, occasionally symptomatic lesions with a very low recurrence rate. These findings reinforce the importance of awareness among clinicians and pathologists to improve diagnostic accuracy and ensure appropriate patient management.

  1. Given that the study is clinically oriented, the manuscript would benefit from a section offering practical diagnostic guidance for clinicians and pathologists. For example, a summary table or diagnostic checklist highlighting key histopathological and clinical features that distinguish oral focal mucinosis from its mimics.

Response: Done

Round 2

Reviewer 2 Report

Comments and Suggestions for Authors

I would like to thank the Authors for their careful revision. The manuscript has improved considerably and is now suitable for publication after minor revisions. Please find my detailed comments and suggestions in the annotated PDF attached.

Author Response

Please see our reply as it is below as word file.

Thank you.
